# Radiographic characteristics of impacted teeth: A retrospective study of 2199 radiograph

**Amin A. Marghalani[1], Arwa U. Alsaggaf[1]\*, Abdullah Hazzazi[2], Anmar Dahlawi[2], Mohd. B. Badaoud[2], Faisal Alhazmi[2], Ibrahim Alnefaie[2], Omair M. Bukhari[3], Abrar K. Demyati[4]**

**1** Faculty of Dentistry, Department of Oral and Maxillofacial Surgery, Division of Prosthodontics, Umm Al-Qura University, Makkah, Saudi Arabia, **2** Faculty of Dental Medicine, Umm Al-Qura University, Makkah, Saudi Arabia, **3** Department of Preventive Dentistry, College of Dental Medicine, Umm Al-Qura University, Makkah, Saudi Arabia, **4** Faculty of Dental Medicine, Department of Oral and Maxillofacial Surgery, Umm Al-Qura University, Makkah, Saudi Arabia

\* ausaggaf@uqu.edu.sa

**Data Availability Statement:** Data cannot be shared publicly because of the confidentiality and ethics.Data are available upon specific request from Umm al-Qura University Dental Hospital

## Abstract

### Objective

To analyze the radiological characteristics of impacted teeth in the population of Makkah in Saudi Arabia.

### Methods

A retrospective analysis was conducted on 2199 digital panoramic radiograph (OPG) scans collected from the database of the dental teaching hospital at Umm Al-Qura University, Makkah, KSA. Out of these, 1503 OPGs (749 males and 754 females) met the inclusion criteria, which required clear OPGs with high quality and good visibility of anatomic structures. The study included patients of both genders, aged between 13 and 70, with complete root formation of impacted teeth.

### Results

The analysis of 1,503 patients' radiographs revealed that 27.70% had impacted third molars, with mandibular molars being more commonly affected than maxillary molars. Additionally, other impacted teeth were observed, including maxillary canines, second premolars, second molars, and supernumerary teeth. Male patients had a slightly higher prevalence of impaction. The majority of impacted third molars were found in the age group of 18-30. The angulation and depth of impacted teeth differed between maxillary and mandibular arches. Most impacted mandibular third molars had a class I relation with the ramus.

### Conclusion

The study provides valuable insights into impacted teeth in Makkah, Saudi Arabia. It highlights a significant occurrence of impacted third molars, as well as other teeth. These

Institutional Data Access / Ethics Committee (contact via akdemyati@uqu.edu.sa or ausaggaf@uqu.edu.sa) for researchers who meet the criteria for access to confidential data.

**Funding:** The author(s) received no specific funding for this work.

**Competing interests:** The authors have declared that no competing interests exist.

findings contribute to a better understanding of impacted teeth and provide important support for dental professionals in their efforts to improve patient outcomes.

## Introduction

Impacted teeth are unerupted teeth that are unable to reach their normal position due to factors such as insufficient space, malposition, or other obstructive elements (American Association of Oral and Maxillofacial Surgeons [1]. While teeth can naturally erupt late, there is a defined time limit for this process. A tooth failing to erupt to its normal position more than a year after its recognized eruption age is classified as impacted [2]. Third molars in the maxilla and mandible are the most commonly impacted teeth, followed by the maxillary canine and mandibular premolars, as they are the last teeth to erupt and are thus more likely to face space constraints [3].

Even though most impactions are considered harmless and can be asymptomatic, complications may arise, causing pain, especially if associated with pathologies such as caries, periodontal diseases, cysts or tumors, pericoronitis, and infections [4]. Rarely, it can lead to severe bone resorption and jaw fractures [5]. Dental caries is a common complication that may affect the impacted lower third molar or the distal surface of the second molar and is considered a common reason for extracting lower third molars [6]. According to Nordenram et al., they found that 15% of third molar extractions are due to caries [7]. Researchers found that older age and erupted third molars were associated with higher rates of occlusal caries in patients with asymptomatic impacted third molars [6]. Another example of how impaction may cause complications is the impaction of the maxillary canine, which increases the risk of cystic follicular lesions, gingival infections, and root resorption of the neighboring lateral incisors [8].

Various classifications of impacted teeth have been established, taking into consideration various factors [9–11]. The classification of impacted mandibular third molars includes considerations of level, position, and depth, as determined by a panoramic radiograph. This classification system includes A) Pell and Gregory radiographic classification with respect to mandibular ramus, categorized into Class I, Class II, and Class III; B) Pell and Gregory radiographic classification based on the occlusal plane, classified as Class A, Class B, and Class C [10]; and C) Winter's classification, which includes mesioangular, distoangular, horizontal, vertical, buccal/lingual obliquity, and transverse categories [9]. For impacted maxillary third molars, the classification includes Class A, Class B, and Class C [12].

Globally, the prevalence of impacted teeth ranges from 16.7% to 68.6%, with third molars and canines found to be the most commonly impacted teeth [13–16]. Canine impaction in the maxilla occurs 20 times more frequently than in the mandible [17]. Several classification systems exist for categorizing impacted maxillary canines [18]. One such classification was developed by Yamamoto et al. and includes a total of seven distinct classes [18]. In addition, it has been discovered that the prevalence of premolar-impacted teeth varies with age. In adults, the reported prevalence is 0.5% [19]. The prevalence of teeth impaction in the local population is crucial as there is evidence of genetic and hereditary factors [20].

Several studies have been conducted in Saudi Arabia to evaluate the prevalence and characteristics of impacted teeth [21–26]. This study aims to analyze the radiological characteristics of impacted teeth in the population of Makkah in Saudi Arabia.

### Clinical significance of the study

The study builds a comprehensive database of impacted teeth which can provide valuable information for future research in the region. Such database can help identify risk factors, and need for proper management, ultimately benefiting both clinical practice and future investigations on impacted teeth.

## Methods

### Ethical approval and consent

This study received approval from the Umm Al-Qura University institutional review board (HAPO-02-k-012–2022-11–1254).

Patients consented with written consent from the first visit to the screening and triage clinic at the dental teaching hospital at Umm Al-Qura University, Makkah, KSA. The written consent indicated that data and radiographs would be used for research purposes and education purposes without identifications. After receiving the ethical approval on 12 November 2022, the dental teaching hospital obtained access to data collection from the medical reports and radiographs of the patients. Permission to access the data for retrospective collection of information started on 15 November 2022. Data collection was done after the examiners' calibration on 22 November 2022.

### Data and sample collection

A retrospective analysis of 2199 digital panoramic radiographs (OPG) were conducted using data from the dental teaching hospital at Umm Al-Qura University, Makkah, KSA, taken between 2020 and 2022. A total of 1503 OPG scans (749 males and 754 females) meeting the inclusion criteria were selected. The age group ranged from 13 to 70 years old. The OPG scans aimed to assess the prevalence of impacted teeth and their radiographic characteristics. Demographic details, such as patient age, gender, and residential information, were obtained from dental records. 377 or more research subjects are needed to have a confidence level of 95% that the real value is within ±5% of the measured value.

### Inclusion criteria

Inclusion criteria included clear OPG with high quality and good visibility of anatomic structures of patients attending the dental teaching hospital at Umm Al-Qura University. Furthermore, the study included patients of both genders, aged between 13 and 70, with complete root formation of impacted teeth. Teeth are considered impacted when they fail to erupt into the oral cavity for more than two years beyond their average eruption time.

### Exclusion criteria

Unclear images, age below 13, one or more pathological situations (endocrinal deficiency such as hypothyroidism, hypopituitarism, trauma, or jaw fractures), and hereditary diseases or syndromes such as Down's syndrome and cleidocranial dysostosis. These patients were excluded because the normal growth of permanent dentition can be affected by these conditions.

### Radiographic analysis

All radiographs were taken by a GENDEX® OPG scanner (GXDP-700TM, Hatfield, USA) by an experienced radiograph technician. Imaging was carried out in the following settings: 70

kV, 16 s, and 71 mGy cm$^2$. All radiographs were analyzed independently by five examiners with the same level of experience in their internship years.

## Examiners and calibration

The radiographs were collected by five dentists during their internship year. Prior to the analysis, 10 radiographs were chosen from the included sample for the determination of the inter-examiner reliability test, and reproducibility was considered good. Inter-examiner reliability data, with a mean percentage agreement of 77.31% (95% CI: 72.7%–82%) and a Fleiss' kappa coefficient of 0.41 (95% CI: 0.31–0.5), demonstrated moderate reliability. The calibration was carried out individually, and a complete analysis of the sample was carried out over a period of one week. The examiners were instructed by a consultant in oral and maxillofacial surgery, who explained the classification of impaction and the recording process. All five calibrated examiners participated in the data collection.

## Classifications

The classifications used to describe the impaction are Pell and Gregory's and Winter's classifications [9,10] (Fig 1) [27].

**Impacted third molars angulation.** According to Winter's classification; Vertical impaction: the long axis of the third molar is parallel to the long axis of the second molar (from 10˚ to 10˚); mesioangular impaction: the impacted tooth is tilted toward the second molar in a mesial direction (from 11˚ to 79˚); horizontal impaction: the long axis of the third molar is

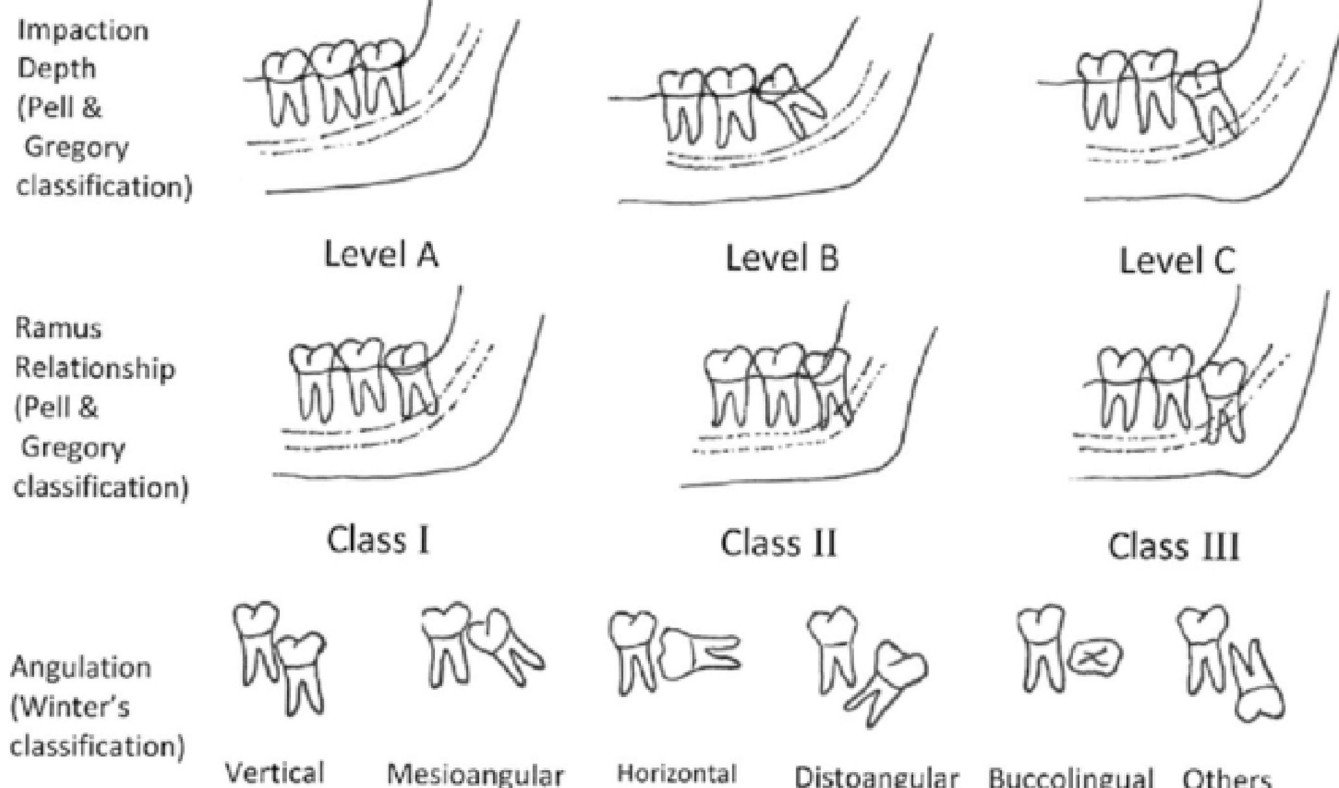

**Fig 1. The classification of Pell and Gregory and the classification of Winter.**

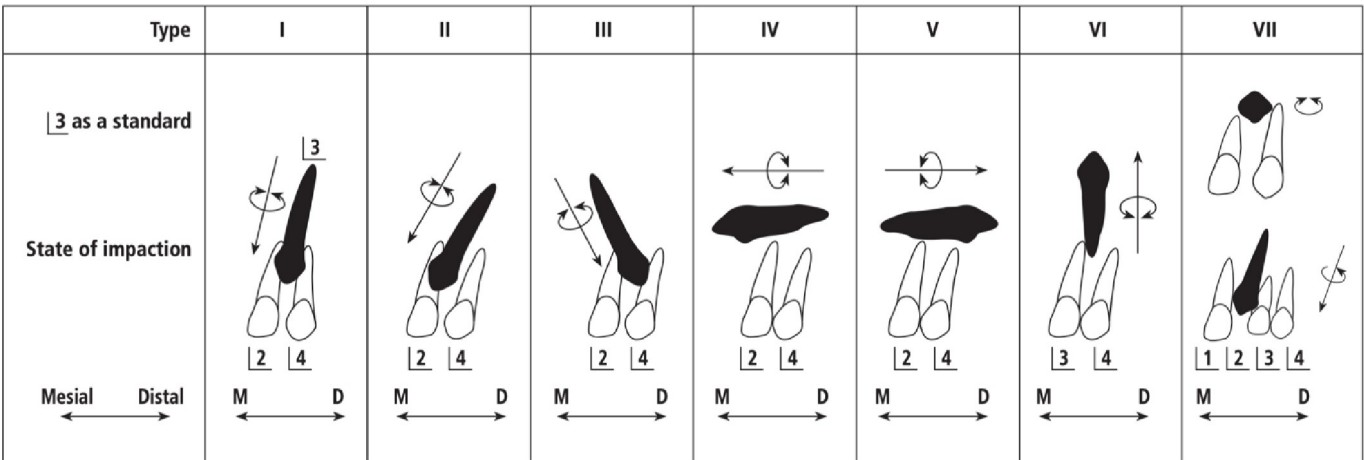

**Fig 2. Classification of Yamamoto et al. impacted maxillary canines.**

horizontal (from 80° to 100°); and distoangular impaction: the impacted tooth is tilted toward the ramus of the mandible in a distal direction (from −11° to −80°).

**Impacted third molars depth.** According to Pell and Gregory's classification; Level A: is when the occlusal surface of the impacted tooth is on the same level as the occlusal plane; Level B: is when the occlusal surface of the impacted tooth is below the occlusal surface and above the cementoenamel junction of the second molar; and Level C: when the occlusal surface of the impacted tooth is below the cementoenamel junction of the second molar.

**Impacted mandibular third molars ramus relationship.** According to Pell and Gregory's classification for impacted mandibular third molars; Class I: sufficient space available between the anterior border of the ascending ramus and distal side of the second molar for third molar eruption. Class II: the space available between the anterior border of the ascending ramus and the distal side of the second molar is less than the mesiodistal width of the third molar's crown. This indicates that the ascending ramus bone covers the distal portion of the third molar crown. Class III: absolute lack of space is observed; the third molar is totally embedded in the ascending ramus bone.

**Impacted maxillary canines' classification.** Impacted maxillary canines can be classified into one of seven types, according to the study by Yamamoto et al. [18] This classification is based on the position of the impacted canines relative to the occlusal plane and their location in relation to the adjacent teeth (Fig 2) [18].

## Statistical analysis

The data was analyzed using STATA version 14. Continuous variables are described using means and standard deviations. Categorical variables are described using frequencies and percentages. A Chi-square test for independence was conducted to examine the association between categorical variables. All Statistical tests' significance levels were 0.05.

## Results

The analysis of the impacted third molars among the 1503 patients' radiographs showed that these molars were impacted in 416 patients (27.70%) with a mean age of 41.5 years. The mean age for females is 27.2 and for males is 29.3. Mandibular third molars showed a higher prevalence of impaction (27.4%) than maxillary third molars (21.49%).

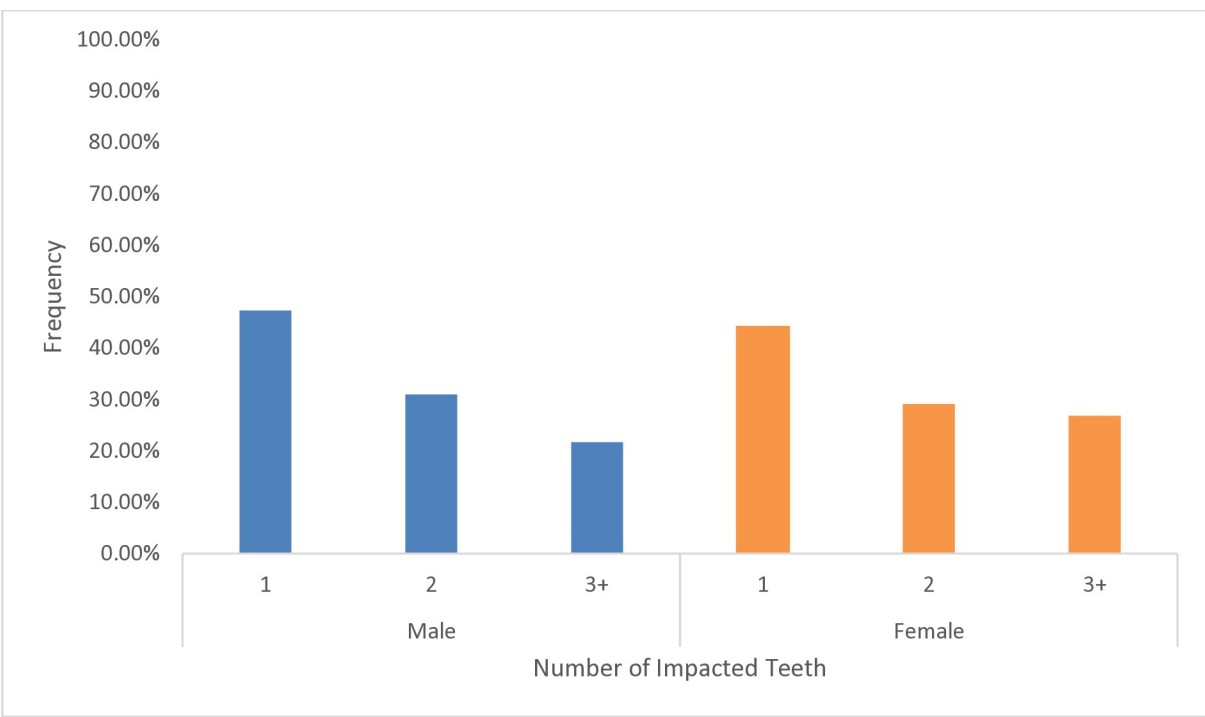

**Fig 3. Distribution of third molar impaction based on the number of impacted teeth.**

The distribution of the impacted third molar according to gender and the number of teeth impacted is shown in (Fig 3). Of all the male patients, 96 (47.3%) had one, 63 (31%) had two, and 44 (21.7%) had three or more teeth impacted. Of all the female patients, 94 (44.1%) had one, 62 (29.1%) had two, and 57 (26.8%) had three or more teeth impacted. The overall impaction was slightly higher among males (51.20%). A digital OPG showing the impacted third molars in all four quadrants is shown in (Fig 4A).

Among all impacted third molars, 67.78% were found in the age group of 18–30. Males represented 49.29%, while females represented 50.71%. The age group of 31–40 had 14.9% of all the impacted third molars, with 51.61% of them being males and 48.39% being females. The age group with the third-most impacted third molars was 41–50 (7.69%), followed by 51–60 (3.84%) and then >60 (1.2%). The distribution of the third molar impaction based on age and gender is shown in (Table 1).

In males, the most prevalent angulation of impaction was vertical in upper right third molars (UR8) (57.14%) and upper left third molars (UL8) (55.84%), while it was horizontal in teeth lower left third molars (LL8) (33.64%) and lower right third molars (LR8) (36.52%). In females, the most prevalent angulation of impaction was also vertical in UR8 (51.22%) and UL8 (45.98%), while it was mesioangular in LL8 (34.34%) and LR8 (34.07%) (Table 2). There was no statistically significant difference between males and females regarding the angulation of impaction except for tooth LR8 ($P < 0.05$).

Most of the impacted third molars' depth of impaction was at level C in the maxillary arch, while it was at level B in the mandibular arch (Table 3). There was no statistically significant difference between males and females regarding depth of impaction except for LR8 ($P < 0.05$).

The most frequent relation of impacted mandibular third molar teeth with the ramus in both males and females was class I, except for LR8 in males, which was class II (Table 4). There

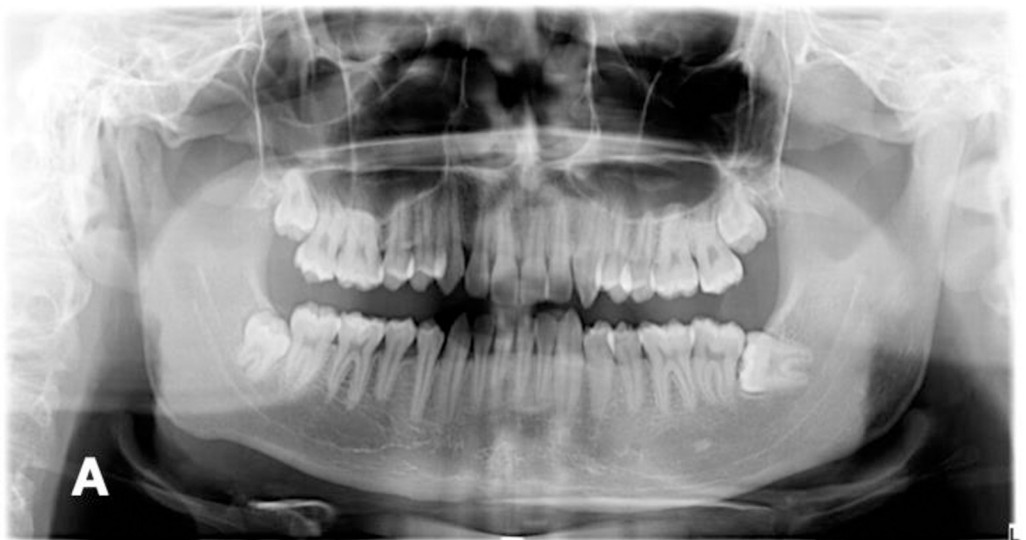

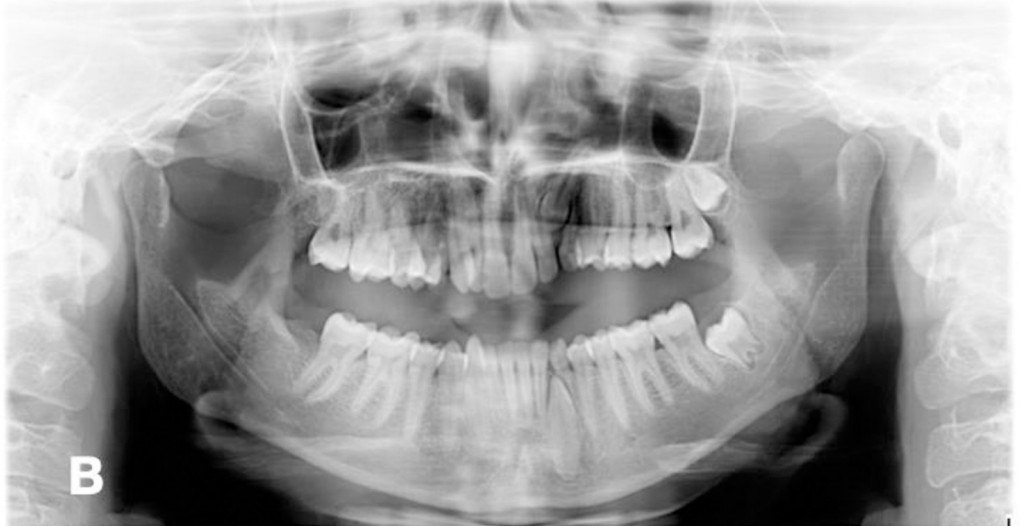

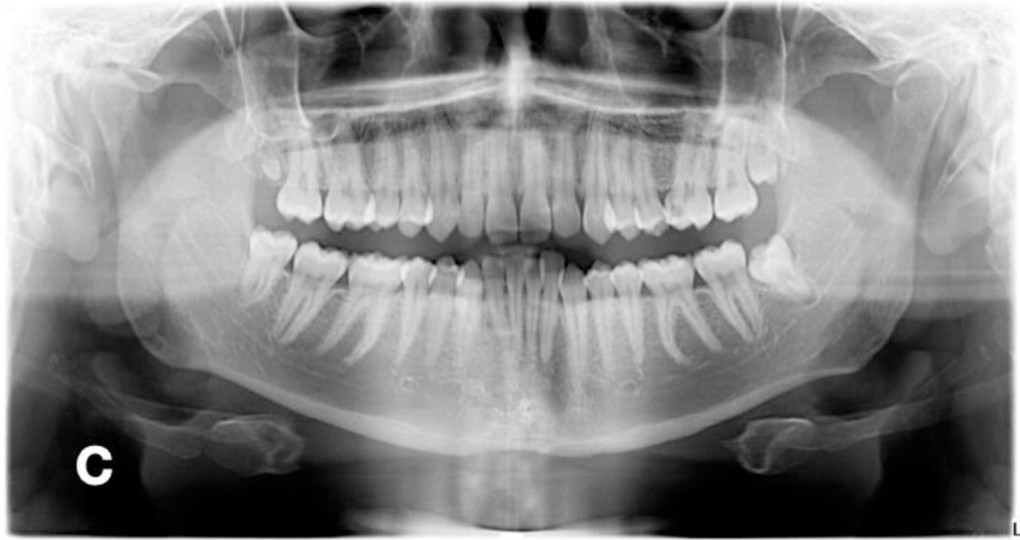

**Fig 4. OPG showing impacted teeth. (A)**: Third molars in all four quadrants. (**B**): Left maxillary and mandibular canines (Type 2). (**C**): Distomolars.

was no statistically significant difference between males and females regarding the relationship of impacted mandibular third molar teeth with the ramus.

Moreover, the most prevalent impacted teeth other than the third molars were canines (n = 37 (2.46%), distributed as 2.4% maxillary canines (n=36) and 0.13% mandibular canines (n=2). Regarding gender, 19 males (2.54%) and 18 females (2.39%) had impacted canines.

Unilateral maxillary impaction was present in 28 subjects (1.86%) [12 (1.59% of females) were females and 16 (2.14% of males) were males], while mandibular unilateral impaction was present in 2 subjects (0.13%) [1 female and 1 male]. However, bilateral maxillary impaction was present in 8 subjects (0.53%) [6 were females (0.8% of females) and 2 were males (0.27% of males)], while mandibular bilateral impaction was NOT present.

The highest rate is Type 2 (65.11%), where the crown of the impacted canine is angled mesially and is covering a portion of the adjacent lateral incisor tooth as seen on the radiograph. It was followed by Type 1 (25.58%), where the impacted canines are vertically and almost perpendicular to the occlusal plane and located between the lateral incisor and first premolar. Then, come the last in our study, type 4 (9.30%), where the impacted maxillary canine is located horizontally and the crown is directed mesially. A digital OPG showing impacted canine in type 2 is shown in (Fig 4B).

The study found that, after third molars and canines, the teeth with the next highest rates of impaction were upper second premolars (0.73%), upper second molars (0.20%), and lower second molars (0.13%). Additionally, the current study also examined the prevalence of distomolars and residences, which were reported as 0.90% and 0.47%, respectively. An example of an impacted distomolar is shown in the OPG in (Fig 4C).

## Discussion

The findings of this retrospective study provide valuable insights into the radiographic characteristics of impacted teeth in the study population of Makkah along with the prevalence, which contribute to the existing knowledge in dental research.

The observed prevalence of impacted teeth in the current study was 27.70% in the study population. This finding differs from previous studies conducted in different populations and regions of Saudi Arabia [21,28,29]. Alamri et al. (2020) reported a prevalence rate of 13.2% for impacted teeth in the eastern region of Saudi Arabia. Idris (2021) reported a prevalence rate of 24.3% for impacted teeth in the Jazan region. Alfadil and Almajed (2020) found a higher prevalence rate of 58.3% in the central region of Saudi Arabia. The variations in prevalence observed

**Table 1. Distribution of third molar impaction based on age and gender.**

| Age | Female | Male | Total |
|---|---|---|---|
| <18 | 10 (2.4%) | 9 (2.16%) | 19 (4.56%) |
| 18-30 | 143 (34.37%) | 139 (33.41%) | 282 (67.78%) |
| 31-40 | 30 (7.21%) | 32 (7.69%) | 62 (14.9%) |
| 41-50 | 16 (3.84%) | 16 (3.84%) | 32 (7.69%) |
| 51-60 | 4 (.96%) | 12 (2.88%) | 16 (3.84%) |
| >60 | 0 | 5 (1.2%) | 5 (1.2%) |
| Total | 203 (48.79%) | 213 (51.2%) | 416 (100%) |

Table 2. Comparison between genders in relation to impacted tooth angulation type, chi-square test (n = 735).

| Gender | Buccolingual | | Distoangular | | Horizontal | | Inverted | | Mesioangular | | Vertical | | Total | P-value |
|---|---|---|---|---|---|---|---|---|---|---|---|---|---|---|
| | No. | % | No. | % | No. | % | No. | % | No. | % | No. | % | No. | |
| Female UR8 | 4 | 4.9 | 25 | 30.5 | 1 | 1.2 | 0 | 0.0 | 10 | 12.2 | 42 | 51.2 | 82 | 0.743 |
| Male UR8 | 1 | 1.3 | 20 | 26.0 | 2 | 2.6 | 0 | 0.0 | 10 | 13.0 | 44 | 57.1 | 77 | |
| Female UL8 | 5 | 5.7 | 24 | 27.6 | 5 | 5.7 | 0 | 0.0 | 13 | 14.9 | 40 | 46.0 | 87 | 0.261 |
| Male UL8 | 4 | 5.2 | 22 | 28.6 | 0 | 0.0 | 0 | 0.0 | 8 | 10.4 | 43 | 55.8 | 77 | |
| Female LL8 | 2 | 2.0 | 3 | 3.0 | 21 | 21.2 | 0 | 0.0 | 34 | 34.3 | 39 | 39.4 | 99 | 0.263 |
| Male LL8 | 4 | 3.7 | 4 | 3.7 | 36 | 33.6 | 1 | 0.9 | 34 | 31.8 | 28 | 26.2 | 107 | |
| Female LR8 | 0 | 0.0 | 8 | 8.8 | 17 | 18.7 | 0 | 0.0 | 31 | 34.1 | 35 | 38.5 | 91 | 0.009 * |
| Male LR8 | 2 | 1.7 | 5 | 4.3 | 42 | 36.5 | 0 | 0.0 | 38 | 33.0 | 28 | 24.3 | 115 | |

among these studies can be attributed to differences in impaction diagnostic criteria, as well as variations in sample sizes and age groups [21,28,29].

The current study investigated the pattern of impacted teeth across various age groups, ranging from 13 to 70 years old, divided into six groups. The findings revealed that the age group of 18–30 years old had the highest prevalence of impacted teeth, accounting for 67.78%. A study found similar results, indicating that the age group of 20–30 years exhibited the highest prevalence of impaction (31.9%), with a subsequent decrease observed in the 30–40 age group [21].

Regarding gender, the prevalence of impaction among females was nearly equal to that among male participants, with females accounting for 50.20% of the cases with impacted teeth.

Third molars are the most impacted teeth, with a worldwide average impaction rate of 24% [30]. The sample in the current study indicated a higher prevalence of mandibular third molars (27.4%) compared to maxillary third molars (21.49%). This finding is consistent with the study conducted by Alfadil and Almajed (2020) in Riyadh, where they reported a prevalence of 58.5% for mandibular third molars and 41.5% for maxillary third molars. The study also reported that the distribution of impacted third molars was approximately equal between the two genders [29].

In addition to the impaction of third molars, the impaction of other teeth is also significant for esthetics and malocclusion. With this in mind, the current study focused on all impacted teeth, rather than being limited to third molars alone. It was reported that the most frequently impacted teeth after the third molars were maxillary permanent canines, accounting for 3.06% of cases. This was followed by upper second premolars (0.73%), upper second molars (0.20%),

Table 3. Comparison between genders in relation to impacted tooth position, chi-square test (n = 735).

| Gender | A | | B | | C | | Total | P-value |
|---|---|---|---|---|---|---|---|---|
| | No. | % | No. | % | No. | % | No. | |
| Female UR8 | 0 | 0.0 | 34 | 41.5 | 48 | 58.5 | 82 | 0.749 |
| Male UR8 | 1 | 1.3 | 32 | 41.6 | 44 | 57.1 | 77 | |
| Female UL8 | 0 | 0.0 | 36 | 41.4 | 51 | 58.6 | 87 | 0.58 |
| Male UL8 | 0 | 0.0 | 36 | 46.8 | 41 | 53.2 | 77 | |
| Female LL8 | 21 | 21.2 | 68 | 68.7 | 10 | 10.1 | 99 | 0.461 |
| Male LL8 | 26 | 24.3 | 64 | 59.8 | 17 | 15.9 | 107 | |
| Female LR8 | 27 | 29.7 | 59 | 64.8 | 5 | 5.5 | 91 | 0.024 * |
| Male LR8 | 32 | 27.8 | 64 | 55.7 | 19 | 16.5 | 115 | |

**Table 4. Comparison between genders in relation to impacted teeth with the ramus, chi-square test (n = 410).**

| Gender | I | | II | | III | | Total | P-value |
|---|---|---|---|---|---|---|---|---|
| | No. | % | No. | % | No. | % | No. | |
| Female LL8 | 56 | 56.0 | 36 | 36.0 | 8 | 8.0 | 100 | 0.815 |
| Male LL8 | 53 | 50.5 | 44 | 41.9 | 8 | 7.6 | 105 | |
| Female LR8 | 43 | 47.3 | 40 | 44.0 | 8 | 8.8 | 91 | 0.178 |
| Male LR8 | 49 | 43.0 | 59 | 51.8 | 6 | 5.3 | 114 | |

and lower second molars (0.13%). Additionally, the current study also examined the prevalence of distomolars and mesiodentes, which were reported as 0.90% and 0.47%, respectively. In Saudi Arabia, several researchers have studied the impaction of teeth other than third molars [31]. One study found that among the impacted teeth, third molars constituted 15.9% of cases, upper canines accounted for 3.3%, lower premolars made up 0.6%, and other teeth, including upper second premolars and lower second premolars, comprised 1.4% of cases [21,31]. In another study, it was reported that the most commonly impacted teeth were maxillary canines (50.4%), followed by upper second premolars (18.2%) and lower second premolars (12.2%) [21].

Given the prevalence of impacted teeth, it is essential to evaluate the position, angulation, and level of impacted teeth with respect to age and gender. This assessment is crucial for improved patient management regarding whether to keep or extract these teeth.

Vertical impaction of the upper third molars was the most frequently observed impaction in both males and females (54.69%). In the mandible, horizontal impaction was the most prevalent type of third molars in males (35.13%), while mesioangular impaction was the most common in females (38.94%). Notably, there was a significant difference between the genders in the mesioangular impaction in the lower right third molars. Similar studies have reported similar findings, demonstrating that vertical angulation is the most frequently observed angulation for impacted upper third molars [28,29,32]. However, when it comes to the angulation of lower third molars, different results have been reported. A study indicated mesioangular impaction as the most common type of angulation of impacted lower molars among the entire sample [29]. A study conducted by Idris et al. (2021) found that the majority of impacted mandibular third molars exhibited vertical angulation (37.10%) [28].

As per the Pell and Gregory classification, in our study, the most frequently observed depth level in the maxilla was level C (56.96%) and in the mandible, it was level B (61.89%). On the other hand, in their study, Kaomongkolgit and Tantanapornkul (2017) found that the most prevalent depth level in both jaws was B (53.86%). In addition, regarding the position of the impacted lower molar relative to the ramus, LL8 exhibited a class I position at 53.17%, while tooth LR8 exhibited a class II position at 48.29%. El-Khateeb et al. (2015) found that among the assessed impacted mandibular third molars, class II position was the most prevalent (89.2%), followed by class I. On the other hand, Alfadil and Almajed (2020) found that the class I position was the most prevalent (66.7%).

A total of 116 individuals experienced the impaction of teeth other than third molars. Regarding the impacted maxillary canines, the majority were positioned in type 2 (65.11%), where the impacted canine is angled mesially, while the remaining canines were oriented vertically as in type 1 (25,58%), and finally type 4 (9.30%) which is perpendicular to the occlusal plane. This result is in accordance with the results of the study conducted by El-Khateeb et al. and a study conducted by Altan et al. [18,32].

This prevalence of impacted teeth in the Makkah region emphasizes the clinical significance and the need for proper management and treatment of impacted teeth to prevent potential

complications and maintain oral health. Impacted teeth can lead to various complications, such as crowding, malocclusion, cyst formation, and damage to adjacent teeth [33]. However, it is important to acknowledge that the current study has limitations in terms of its sample selection, as it focused exclusively on the population attending a single hospital, specifically the dental teaching hospital in Makkah. This restriction limits the generalizability of the findings to other regions or populations. Additionally, as a cross-sectional study, our findings are limited in making causal inferences. While digital OPGs provide reasonably accurate results, their effectiveness is hindered by challenges such as magnification, superimposition, and distortion. Therefore, it is recommended that future studies be conducted with a diverse range of populations in Saudi Arabia to ensure greater accuracy in generalizing the results.

## Conclusion

The current study has provided valuable insights into the characteristics of impacted teeth. The findings demonstrate that a significant portion of the population in Makkah, Saudi Arabia, experiences impacted teeth, with mandibular third molars, maxillary third molars, and maxillary canines being the most commonly impacted teeth. These findings enhance the understanding of impacted teeth and provide valuable support for dental professionals in terms of diagnosis, treatment planning, and patient management. Further research and clinical interventions are necessary to effectively address the challenges associated with impacted teeth.

## Author Contributions

**Conceptualization:** Amin A. Marghalani, Arwa U. Alsaggaf, Omair M. Bukhari, Abrar K. Demyati.

**Formal analysis:** Omair M. Bukhari.

**Investigation:** Amin A. Marghalani, Abdullah Hazzazi, Anmar Dahlawi, Mohd. B. Badaoud, Faisal Alhazmi, Ibrahim Alnefaie, Abrar K. Demyati.

**Methodology:** Amin A. Marghalani, Arwa U. Alsaggaf, Abdullah Hazzazi, Anmar Dahlawi, Mohd. B. Badaoud, Faisal Alhazmi, Ibrahim Alnefaie, Omair M. Bukhari, Abrar K. Demyati.

**Software:** Abdullah Hazzazi, Anmar Dahlawi, Mohd. B. Badaoud, Faisal Alhazmi, Ibrahim Alnefaie, Omair M. Bukhari, Abrar K. Demyati.

**Supervision:** Amin A. Marghalani, Arwa U. Alsaggaf, Omair M. Bukhari, Abrar K. Demyati.

**Validation:** Amin A. Marghalani, Arwa U. Alsaggaf, Abdullah Hazzazi, Anmar Dahlawi, Mohd. B. Badaoud, Faisal Alhazmi, Ibrahim Alnefaie, Abrar K. Demyati.

**Visualization:** Arwa U. Alsaggaf.

**Writing – original draft:** Amin A. Marghalani, Arwa U. Alsaggaf, Abdullah Hazzazi, Anmar Dahlawi, Mohd. B. Badaoud, Faisal Alhazmi, Ibrahim Alnefaie, Abrar K. Demyati.

**Writing – review & editing:** Amin A. Marghalani, Arwa U. Alsaggaf, Abdullah Hazzazi, Anmar Dahlawi, Mohd. B. Badaoud, Faisal Alhazmi, Omair M. Bukhari, Abrar K. Demyati.

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
