## [Decision Letter · Decision Letter 0]

23 Jul 2024

PONE-D-24-15648Radiographic Characteristics of Impacted Teeth: A Retrospective Study of 2199 RadiographPLOS ONE

Dear Dr. Alsaggaf,

Thank you for submitting your manuscript to PLOS ONE. After careful consideration, we feel that it has merit but does not fully meet PLOS ONE’s publication criteria as it currently stands. Therefore, we invite you to submit a revised version of the manuscript that addresses the points raised during the review process.

We look forward to receiving your revised manuscript.

Kind regards,

Kumar Chandan Srivastava, BDS, MDS, PhD, MFD RSCI, MFDS RCPS, MFDS RCSEd MDT

Academic Editor

PLOS ONE

Journal Requirements:

"The authors declare that they have no known competing financial interests or personal relationships that could have appeared to influence the work reported in this paper."

3. In the online submission form, you indicated that [Data are available upon specific request from the

Umm al-Qura University Dental Hospital Institutional Data Access / Ethics Committee for researchers who meet the criteria for access to confidential data. Data cannot be shared publicly because of the confidentiality and ethics.]. 

Additional Editor Comments:

Dear Authors,

Kindly carefully go through the reviewer comments and revise the manuscript accordingly.

Best Wishes

Reviewers' comments:

Reviewer's Responses to Questions

**Comments to the Author**

1. Is the manuscript technically sound, and do the data support the conclusions?

Reviewer #1: Partly

Reviewer #2: Yes

2. Has the statistical analysis been performed appropriately and rigorously? 

Reviewer #1: Yes

Reviewer #2: Yes

3. Have the authors made all data underlying the findings in their manuscript fully available?

Reviewer #1: Yes

Reviewer #2: Yes

4. Is the manuscript presented in an intelligible fashion and written in standard English?

Reviewer #1: Yes

Reviewer #2: Yes

5. Review Comments to the Author

Reviewer #1: Mention the power of the study

How was the sample size calculated

Mention the clinical significance of the study

Mention the mean age of Males and females separately

Grammatical errors need to be addressed

Reviewer #2: Thank you for entrusting me with the evaluation of your manuscript. Your study, "Radiographic Characteristics of Impacted Teeth: A Retrospective Study of 2199 Radiograph" is a significant contribution to our field.

I found your paper on the radiographic characteristics of impacted teeth to be quite intriguing. However, I do have some points that I believe could further enhance your work.

In the second paragraph of the introduction, complications that impacted teeth may be shared in more detail, and rates can be given. The authors can benefit from the following article on this subject.

Altan A, Akbulut N. Does the Angulation of an Impacted Mandibular Third Molar Affect the Prevalence of Preoperative Pathoses? J Dent (Shiraz). 2019 Mar;20(1):48-52. doi: 10.30476/DENTJODS.2019.44563. PMID: 30937337; PMCID: PMC6421329.

The article gives information about the radiographic characteristics of impacted teeth. Classifications such as Winter and Pell Gregory have been made for impacted wisdom teeth. It has been stated that the most frequently impacted tooth after impacted wisdom teeth is the maxillary canine teeth. There are some classifications regarding maxillary canine teeth. One of them is mentioned in the article below. It is recommended that information about the classifications of maxillary canines be provided.

Altan, A., Çolak, S., Akbulut, N., & Altan, H. (2020). Radiographic features and treatment strategies of impacted maxillary canines. Cumhuriyet Dental Journal, 23(1), 32-37.

Best regards

6. PLOS authors have the option to publish the peer review history of their article (what does this mean?). If published, this will include your full peer review and any attached files.

Reviewer #1: No

Reviewer #2: No

---

## [Author Response · Author response to Decision Letter 0]

13 Aug 2024

For Reviewer #1: 

Thank you for your comments. Please find the response for each point, beside highlighting it in the revised version of the manuscript.

1) Mention the power of the study. How was the sample size calculated.

The required sample size was calculated for prevalence and therefore is reported in this format with confidence level rather than power.

Added to the text (Material and methods). 377 or more research subjects are needed to have a confidence level of 95% that the real value is within ±5% of the measured value.

2) Mention the clinical significance of the study.

Added to the text (introduction).

3) Mention the mean age of Males and females separately.

Added to the results.

Mean age for female is 27.2 & mean age for male is 29.3. 

4) Grammatical errors need to be addressed. 

Done

For Reviewer #2: 

Thank you for your valid opinion. We appreciated your suggestion. We considered the impacted canine in more details in our paper (highlighted) and we used the reference provided. We elaborated more in the result about the impacted canine. We have also updated the introduction as suggested. (highlighted)

---

## [Decision Letter · Decision Letter 1]

17 Dec 2024

Radiographic Characteristics of Impacted Teeth: A Retrospective Study of 2199 Radiograph

PONE-D-24-15648R1

Dear Dr. Alsaggaf,

We’re pleased to inform you that your manuscript has been judged scientifically suitable for publication and will be formally accepted for publication once it meets all outstanding technical requirements.

Kind regards,

Alexander Maniangat Luke, PhD

Academic Editor

PLOS ONE

Additional Editor Comments (optional):

Reviewers' comments:

Reviewer's Responses to Questions

**Comments to the Author**

1. If the authors have adequately addressed your comments raised in a previous round of review and you feel that this manuscript is now acceptable for publication, you may indicate that here to bypass the “Comments to the Author” section, enter your conflict of interest statement in the “Confidential to Editor” section, and submit your "Accept" recommendation.

Reviewer #1: All comments have been addressed

Reviewer #2: All comments have been addressed

2. Is the manuscript technically sound, and do the data support the conclusions?

Reviewer #1: Partly

Reviewer #2: Yes

3. Has the statistical analysis been performed appropriately and rigorously? 

Reviewer #1: Yes

Reviewer #2: Yes

4. Have the authors made all data underlying the findings in their manuscript fully available?

Reviewer #1: No

Reviewer #2: Yes

5. Is the manuscript presented in an intelligible fashion and written in standard English?

Reviewer #1: Yes

Reviewer #2: Yes

6. Review Comments to the Author

Reviewer #1: (No Response)

Reviewer #2: Thank you for entrusting me with the evaluation of your manuscript. The authors have made the requested changes. After these changes, the article can be accepted.

7. PLOS authors have the option to publish the peer review history of their article (what does this mean?). If published, this will include your full peer review and any attached files.

Reviewer #1: No

Reviewer #2: No

---

## [Editor Report · Acceptance letter]

14 Jan 2025

PONE-D-24-15648R1 

PLOS ONE

Dear Dr. Alsaggaf, 

I'm pleased to inform you that your manuscript has been deemed suitable for publication in PLOS ONE. Congratulations! Your manuscript is now being handed over to our production team.

Kind regards, 

on behalf of

Dr. Alexander Maniangat Luke 

Academic Editor

PLOS ONE